# A bend, flip and trap mechanism for transposon integration

**Elizabeth R Morris[1†], Heather Grey[1], Grant McKenzie[2], Anita C Jones[2], Julia M Richardson[1]***

[1]Institute of Quantitative Biology, Biochemistry and Biotechnology, University of Edinburgh, Edinburgh, United Kingdom; [2]EaStCHEM School of Chemistry, Edinburgh, United Kingdom

**Abstract** Cut-and-paste DNA transposons of the *mariner/Tc1* family are useful tools for genome engineering and are inserted specifically at TA target sites. A crystal structure of the mariner transposase Mos1 (derived from *Drosophila mauritiana*), in complex with transposon ends covalently joined to target DNA, portrays the transposition machinery after DNA integration. It reveals severe distortion of target DNA and flipping of the target adenines into extra-helical positions. Fluorescence experiments confirm dynamic base flipping in solution. Transposase residues W159, R186, F187 and K190 stabilise the target DNA distortions and are required for efficient transposon integration and transposition in vitro. Transposase recognises the flipped target adenines via base-specific interactions with backbone atoms, offering a molecular basis for TA target sequence selection. Our results will provide a template for re-designing mariner/Tc1 transposases with modified target specificities.

**\*For correspondence:** jrichard@ staffmail.ed.ac.uk

**Present address:** [†]Mill Hill Laboratory, Francis Crick Institute, London, United Kingdom

**Competing interests:** The authors declare that no competing interests exist.

## Introduction

Transposable elements are ubiquitous in most genomes and promote evolution by generating genetic diversity (*Biémont and Vieira, 2006*). Invading transposons can alter genes, affect gene expression or spread drug resistance in bacteria. As shuffling of DNA by transposition can be deleterious, transposons often become inactivated or transcriptionally silenced. Conversely, transposons can give rise to new, useful cell functions. For example, domestication of a Transib-type DNA transposon led to V(D)J recombination and adaptive immunity in jawed vertebrates (*Kapitonov and Jurka, 2005*). Similarly, the Cas1 integrase component of prokaryotic CRISPR-Cas systems of adaptive immunity originated from DNA transposons named Casposons (*Krupovic et al., 2014*). Integration of spacer sequences into the CRISPR locus by the Cas1-Cas2 complex has similarities with transposon and retroviral DNA integration (*Nuñez et al., 2015*).

DNA transposons move from one genomic location to another using transposon-encoded recombinases, often by a DNA cut-and-paste mechanism. Many DNA transposases (e.g. Mos1, Tn5 and bacteriophage MuA) share a conserved RNase H-like catalytic domain, along with retroviral integrases (e.g. HIV-1) and RAG recombinases. These DDE/D enzymes use common active site chemistry to perform similar DNA cleavage and DNA integration reactions. By contrast, there is wide diversity in their preferred target integration sites. Most DDE/D recombinases show only limited preference for a consensus target DNA sequence, which is usually palindromic (*Goryshin et al., 1998*; *Halling and Kleckner, 1982*). The number of base pairs separating the integration sites on complementary DNA strands also varies, from 2 to 9. Some retroviral integrases (e.g. prototype foamy virus (PFV) and HIV-1) preferentially insert their viral genome into nucleosomal DNA (*Pruss et al., 1994*; *Maskell et al., 2015*). Similarly, some transposases (e.g. Tn10) favour bent target DNA structures (*Pribil and Haniford, 2003*). In other transposition systems (e.g. IS*21*, Mu), an element-encoded

**eLife digest** The complete set of DNA in a cell is referred to as its genome. Most genomes contain short fragments of DNA called transposons that can jump from one place to another. Transposons carry sections of DNA with them when they move, which creates diversity and can influence the evolution of a species. Transposons are also being exploited to develop tools for biotechnology and medical applications. One family of transposons – the Mariner/Tc1 family – has proved particularly useful in these endeavours because it is widespread in nature and can jump around the genomes of a broad range of species, including mammals.

DNA transposons are cut out of their position and then pasted at a new site by an enzyme called transposase, which is encoded by some of the DNA within the transposon. DNA is made up of strings of molecules called bases and Mariner/Tc1-family transposons can only insert into a new position in the genome at sites that have a specific sequence of two bases. However, it was not known how this target sequence is chosen and how the transposon inserts into it.

Morris et al. have now used a technique called X-ray crystallography to build a three-dimensional model of a Mariner/Tc1-family transposon as it inserts into a new position. The model shows that, as the transposon is pasted into its new site, the surrounding DNA bends. This causes two DNA bases in the surrounding DNA to flip out from their normal position in the DNA molecule, which enables them to be recognised by the transposase. Further experiments showed that this base-flipping is dynamic, that is, the two bases continuously flip in and out of position. Furthermore, Morris et al. identified which parts of the transposase enzyme are required for the transposon to be efficiently pasted into the genome.

Together these findings may help researchers to alter the transposase so that it can insert the transposon into different locations in a genome. This will hopefully lead to new tools for biotechnology and medical applications.

accessory ATPase facilitates strand transfer (*Mizuno et al., 2013*; *Arias-Palomo and Berger, 2015*); and can prevent self-destructive insertion of the transposon into its own sequence (target immunity), (*Mizuno et al., 2013*). Despite this biochemical knowledge, the molecular and structural origins for transposon target specificities remain unknown.

Mariner/Tc1/IS630 family transposases are unusual as they integrate their transposons, with a 2 bp stagger, strictly at TA target sequences (*Tellier et al., 2015*). They are widespread in nature and are used as tools for genome engineering and therapeutic applications. For example, the reconstructed Tc1 transposase Sleeping Beauty (*Ivics et al., 1997*) is being used in human clinical trials to treat B-cell lymphoma by genetic engineering of T cells (*Maiti et al., 2013*) and in pre-clinical studies to reduce age-related macular degeneration (*Johnen et al., 2012*). Up to 45 kb of DNA can be inserted into the *C. elegans* genome using a transposition system engineered from the *mariner* transposon Mos1 from *Drosophila mauritiana* (*Frøkjær-Jensen et al., 2014*). The ability to pre-select specific sites for integration, beyond the requisite TA, may be desirable for certain genome engineering applications, e.g. controlled genomic integration of a therapeutic gene. Such targeted transposition has been achieved for the bacterial transposase ISY100 using a C-terminal Zif268 DNA-binding domain fusion (*Feng et al., 2010*); and for Sleeping Beauty transposase either by fusing it with a targeting domain (*Yant et al., 2007*) or by exploiting interactions with a targeting protein (*Ivics et al., 2007*). Conversely, it may be useful to randomise *mariner/Tc1* integrations; for example in whole genome sequencing applications as an alternative to Tn5 (*Amini et al., 2014*). Understanding in molecular detail how *mariner/Tc1* transposons are integrated at TA target sites will aid development of these elements as genome engineering tools.

The wealth of structural and biochemical data for the naturally active, eukaryotic transposon *Mos1* offers a paradigm for determining the molecular mechanism of *mariner/Tc1* transposon integration. The 1286 bp transposon is framed by 28 bp imperfect inverted repeats (IR) (*Jacobson et al., 1986*) and encodes a 345 amino acid transposase that can perform cut-and-paste DNA transposition in vitro (*Lampe et al., 1996*), as shown in *Figure 1a*. The Mos1 transposase homodimer binds to the IR at one transposon end (*Cuypers et al., 2013*) and then captures the

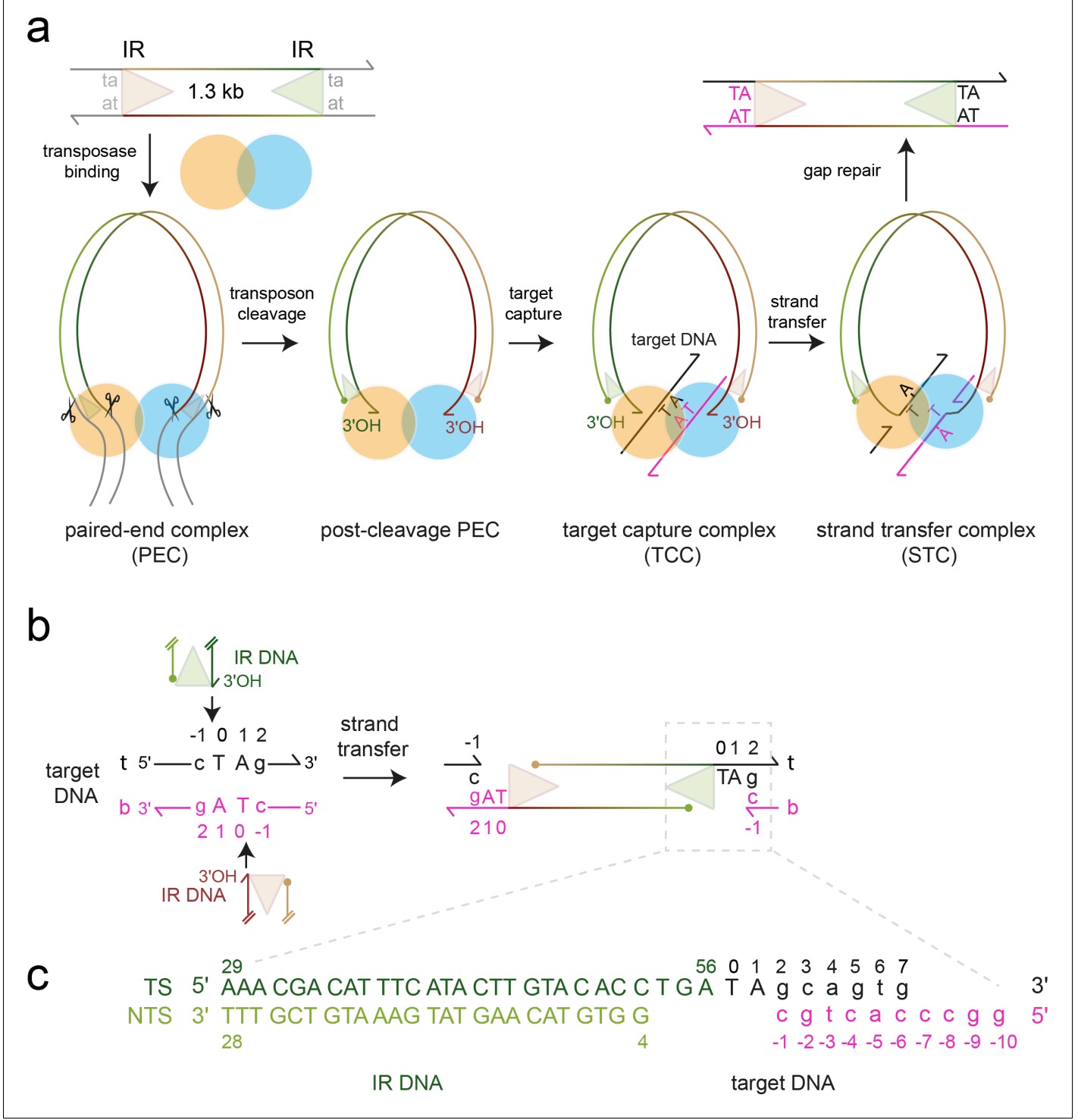

**Figure 1.** Mos1 transposition. (**a**) Schematic of pathway and complexes formed. Each transposon end has a 28 bp IR sequence (triangle) flanked by the TA target site duplication. First and second strand cleavages (scissors) are staggered by three bp and generate a 5' phosphate (filled circle) on the non-transferred strand (NTS), 3 bases within the IR, and a 3'OH (arrow) at the transferred strand (TS) end, respectively. After target DNA capture, the transposon 3' ends integrate at a symmetrical TA sequence, resulting in a 5 nt gap. Gap repair duplicates the TA. (**b**) Mos1 strand transfer. The transposon 3'OHs attack the phosphodiester bond between $T_0$ and $C_{-1}$ on both the top (t, black) and bottom (b, magenta) target DNA strands, joining each TS to target DNA, separating the TA base pairs, and leaving a 3'OH at $C_{-1}$. (**c**) Sequence and numbering of the DNA used to crystallise the STC; see also **Table 1**.

other IR, forming a paired-end complex (PEC). The *trans* architecture of the PEC regulates coordinated excision of the transposon ends (*Richardson et al., 2009*) and cross-talk between transposase sub-units (*Bouuaert et al., 2014*; *Dornan et al., 2015*). After excision, the Mos1 transpososome locates a TA target integration site (*Pflieger et al., 2014*) and, upon binding, forms a target capture complex (TCC) (*Figure 1a*). Attack by the 3'-OH at each transposon end on the phosphodiester 5' of the TA dinucleotide joins the excised transposon to the target site, in the DNA strand transfer reaction (*Figure 1a,b*). The DNA product of transposition, which contains a gap at each transposon end, is bound to the transposase in a strand transfer complex (STC).

To establish how mariner/Tc1 transposases recognize and integrate transposon DNA at a TA dinucleotide, we determined a crystal structure of the Mos1 STC. In this snapshot of the transposition machinery after DNA transposon integration, the target DNA is distorted from B-form and the target adenines are flipped into extra-helical positions. Base-flipping is confirmed in solution by time-resolved fluorescence of strand transfer complexes in which the target adenines are substituted by 2-aminopurine. Adenine-specific interactions, between the flipped adenine bases and transposase backbone atoms, provide a molecular basis for recognition of the TA target sequence. Interactions with Mos1 transposase residues W159, R186, F187 and K190, which are essential for transposon integration in vitro, stabilise distortions in the target DNA. Conservation of key residues involved in stabilising the target DNA distortions suggests this mechanism may also occur with other *mariner/Tc1* family transposons.

**Table 1.** Sequences of oligonucleotides used in the crystallisation, target integration and fluorescence experiments. The target TA dinucleotide (and its variants) are highlighted in bold. The adenine analogue 2-aminopurine is denoted P and 2,6-diaminopurine is D; the thymine analogue 2-thio-thymine, is indicated by S. IR700 indicates the 5' addition of the infrared fluorescent dye 700.

| Name | Sequence | Length (nt) |
| --- | --- | --- |
| Crystallisation of STC | | |
| TS | 5' AAA CGA CAT TTC ATA CTT GTA CAC CTG A**TA** GCA GTG | 36 |
| NTS | 5' GGT GTA CAA GTA TGA AAT GTC GTT T | 25 |
| target DNA | 5' GGC CCA CTG C | 10 |
| Target Integration Assays | | |
| TS IR DNA | 5' AAA CGA CAT TTC ATA CTT GTA CAC CTG A | 28 |
| TS 5' labelled IR DNA | 5' IR700 / AAA CGA CAT TTC ATA CTT GTA CAC CTG A | 28 |
| NTS IR DNA | 5' GGT GTA CAA GTA TGA AAT GTC GTT T | 25 |
| TA target DNA (top strand) | 5' AGC AGT GCA C**TA** GTG CAC GAC CGT TCA AAG CTT CGG AAC GGG ACA CTG TT | 50 |
| TA target DNA (bottom strand) | 5' AAC AGT GTC CCG TTC CGA AGC TTT GAA CGG TCG TGC AC**T A**GT GCA CTG CT | 50 |
| TP target DNA (top strand) | 5' AGC AGT GCA C**TP** GTG CAC GAC CGT TCA AAG CTT CGG AAC GGG ACA CTG TT | 50 |
| TP target DNA (bottom strand) | 5' AAC AGT GTC CCG TTC CGA AGC TTT GAA CGG TCG TGC AC**T P**GT GCA CTG CT | 50 |
| TD target DNA (top strand) | 5' AGC AGT GCA C**TD** GTG CAC GAC CGT TCA AAG CTT CGG AAC GGG ACA CTG TT | 50 |
| TD target DNA (bottom strand) | 5' AAC AGT GTC CCG TTC CGA AGC TTT GAA CGG TCG TGC AC**T D**GT GCA CTG CT | 50 |
| SD target DNA (top strand) | 5' AGC AGT GCA C**SD** GTG CAC GAC CGT TCA AAG CTT CGG AAC GGG ACA CTG TT | 50 |
| SD target DNA (bottom strand) | 5' AAC AGT GTC CCG TTC CGA AGC TTT GAA CGG TCG TGC AC**S D**GT GCA CTG CT | 50 |
| Fluorescence experiments | | |
| TS_P1 | 5' AAA CGA CAT TTC ATA CTT GTA CAC CTG At**P** gca gtg gac gta ggc c | 46 |
| TS_P13 | 5' AAA CGA CAT TTC ATA CTT GTA CAC CTG Ata gca gtg gac gt**P** ggc c | 46 |
| TS_A1 | 5' AAA CGA CAT TTC ATA CTT GTA CAC CTG At**a** gca gtg gac gta ggc c | 46 |
| NTS | 5' GGT GTA CAA GTA TGA AAT GTC GTT T | 25 |
| Target_16 | 5' g gcc tac gtc cac tgc | 16 |

Morris *et al.* eLife 2016;5:e15537. DOI: 10.7554/eLife.15537

**Table 2.** X-ray diffraction and refinement statistics.

| Crystal | Mos1 Strand transfer complex | |
|---|---|---|
| PDB ID | 5HOO | |
| Space group | C121 | |
| Cell dimensions | a = 256.3 Å b = 58.9 Å c = 110.2 Å<br>$\alpha$ = 90.0°, $\beta$ = 94.9°, $\gamma$ = 90.0° | |
| Wavelength (Å) | 0.9795 | |
| Average mosaicity | 0.22 | |
| | *Overall* | *Outer shell* |
| Resolution (Å) | 86.99–3.29 | 3.52–3.29 |
| $R_{symm}$ | 0.077 | 0.152 |
| Total observations | 78358 | 14630 |
| Unique observations | 25201 | 4479 |
| $< I>/\sigma<I>$ | 8.1 | 3.3 |
| Correlation CC | 0.927 | 0.996 |
| Completeness (%) | 99.6 | 99.5 |
| Multiplicity | 3.1 | 3.3 |
| $R_{work}$ | 0.243 | |
| $R_{free}$ (5.21% of reflections) | 0.279 | |
| R.m.s. deviations:<br>Bond Length (Å)<br>Bond Angle (deg)<br>Chiral volume (Å) | 0.0077<br>1.2072<br>0.0785 | |
| Average B factor (Å$^2$) | 74.0 | |
| Ramachandran plot: Core (%)<br>Allowed (%)<br>Outliers (%) | 90.8<br>9.2<br>0 | |

## Results

### Crystallisation of the Mos1 strand transfer complex

To assemble the Mos1 STC, full length T216A Mos1 transposase was combined, in a 1:1 molar ratio, with DNA representing the product of transposon integration (*Figure 1c*). This DNA contains the transposon IR joined at its 3' end to an unpaired TA dinucleotide and target DNA (*Table 1*). The bottom target DNA strand (strand b, magenta, *Figure 1c*) has a cohesive 4 nt 5' overhang (sequence GGCC) to promote interactions between adjacent complexes in the crystal lattice. This approach, of assembling the STC using the strand transfer product, and bypassing catalysis of integration, proved successful for the preparation of *bona fide* PFV strand transfer complexes (*Yin et al., 2012*).

Mos1 STC crystals diffracted X-rays to a maximum resolution of 3.3 Å. Crystallographic phases were determined by molecular replacement (Materials and methods). The difference electron density after molecular replacement and before model building is shown in *Figure 2—figure supplement 1*. The crystallographic asymmetric unit contains one Mos1 STC and, as predicted, base pairing of the 4 nt overhangs in adjacent complexes facilitates crystal packing (*Figure 2—figure supplement 2*). The refined model has an R(free) of 27.9% and good stereochemistry. The X-ray diffraction and refinement statistics are shown in *Table 2*.

### Architecture of the Mos1 strand transfer complex

The refined Mos1 STC crystal structure (*Figure 2a*) contains a transposase homodimer bound to two DNA duplexes representing the products of transposon integration. Target DNA binds in a channel between the two catalytic domains and the active sites contain the strand transfer products. As the

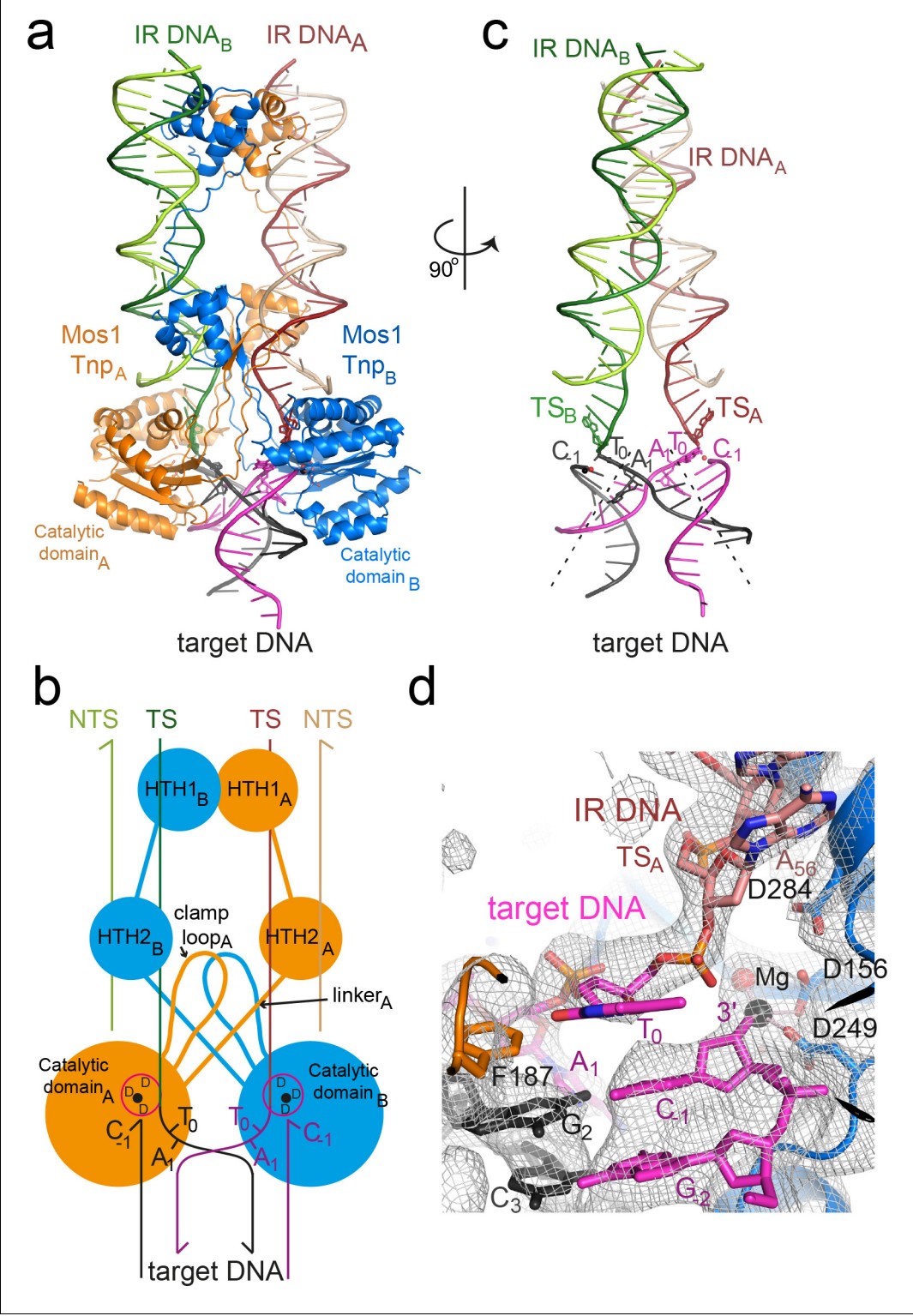

**Figure 2.** Architecture of the Mos1 strand transfer complex. (**a**) Structure of the STC, with transposase subunits (orange and blue), IR DNA (orange and green) and target DNA (magenta and black). *Figure 2—figure supplement 1* shows the crystal packing arrangement. (**b**) Schematic of the Mos1 STC. See *Figure 2—figure supplement 2* for details of transposase DNA interactions. (**c**) DNA components of the STC: target DNA is bent and each IR TS connects at the 3' end to a target DNA strand. (**d**) The active site of catalytic domain B, showing the product of strand transfer into the bottom target strand (magenta). The simulated annealing composite omit

*Figure 2 continued on next page*

*Figure 2 continued*

2Fo-Fc electron density map (grey mesh) is contoured at 1.2σ. The single $Mg^{2+}$ is coordinated by D249, D156 and the 3'OH of $C_{-1}$.

The following figure supplements are available for figure 2:

**Figure supplement 1.** Stereo views of the difference electron density after molecular replacement.

**Figure supplement 2.** Packing arrangement and DNA interactions in the Mos1 STC crystal lattice.

**Figure supplement 3.** Schematic depiction of the interactions between transposase and DNA in the Mos1 STC structure.

TCC also contains a transposase dimer (*Pflieger et al., 2014*), our new STC structure indicates that Mos1 strand transfer, like transposon excision, is catalysed by a transposase dimer.

The transposase subunits adopt a crossed (or *trans*) arrangement with IR DNA in the Mos1 STC, similar to the *pre-* and *post-*TS cleavage Mos1 PECs (*Dornan et al., 2015*; *Richardson et al., 2009*): each IR is recognised by the DNA-binding domain of one transposase subunit and by the catalytic domain of the other subunit (*Figure 2b*), and *vice versa*. This brings the cleaved transposon ends together, and ensures suitable spacing for their integration into TA target DNA. Transposase interactions with IR DNA in the STC (*Figure 2—figure supplement 3*) are similar to those in the *post-*TS cleavage PEC, suggesting that, after transposon excision, the transpososome is poised for target capture. Thus, rather than rearrange the transposase and IR DNA, target DNA is deformed to enable strand transfer.

## Mos1 transposase sharply bends target DNA

The target DNA is severely distorted from B-form conformation (*Figure 2c*): the backbone is bent by 147°, with the apex of the bend at the TA target dinucleotide. DNA unwinds most readily at TA sequences due to the inherent bendability of a pyrimidine-purine step (which has minimal base-to-base overlap and low twist angles) and the lower stability of a TA base pair, compared to CG. The strand transfer reaction creates a nick in both target DNA strands 5' of the target thymine $T_0$, which likely relieves steric constraints and allows the extreme bend across the TA di-nucleotide. Transposase interactions with the backbone phosphates of target nucleotides surrounding the TA sequence support this conformation (*Figure 2—figure supplement 3*).

## Strand transfer products are in proximity to the active sites

The transposase performs three nucleophilic substitution reactions at each transposon end: sequential hydrolysis of both DNA strands to excise the transposon, followed by strand transfer to join the IR to target DNA (*Figure 1a*). One IR is transferred to the top strand (t, black, *Figure 1b*), and the other to the bottom strand (b, magenta, *Figure 1b*). In-line $S_N2$ attack by each transposon 3'-OH on the scissile $T_0$ target DNA phosphate (*Figure 1c*) creates a new bond between the transposon end and the target thymine ($T_0$). At the same time, the phosphodiester linking $C_{-1}$ and $T_0$ is broken, leaving a 3'-OH on $C_{-1}$ and inverting the stereochemistry of the scissile $T_0$ phosphate.

Each Mos1 transposase active site comprises the carboxylate side-chains of three conserved aspartates (D156, D249 and D284) from the same catalytic domain, which coordinate the metal ions ($Mg^{2+}$ or $Mn^{2+}$) required for catalysis. One $Mg^{2+}$ was observed in each active site in the Mos1 STC, coordinated by the D156 and D249 carboxylates, the 3'-OH of $C_{-1}$ and a water molecule. The phosphodiester joining each transposon 3' end ($A_{56}$) to a $T_0$ passes close to an active site (*Figure 2b,c,d*). The $T_0$ phosphate oxygens are 4.4 Å and 7 Å from the $Mg^{2+}$, precluding chelation. Moreover, the $C_{-1}$ 3'-OH is not in-line with the $T_0$–$A_{56}$ phosphodiester bond, consistent with repositioning of the nascent transposon-target DNA junction, away from the active site $Mg^{2+}$ after strand transfer. Similar to the PFV STC (*Maertens et al., 2010*), this likely prevents self-destructive disintegration and drives transposition forwards.

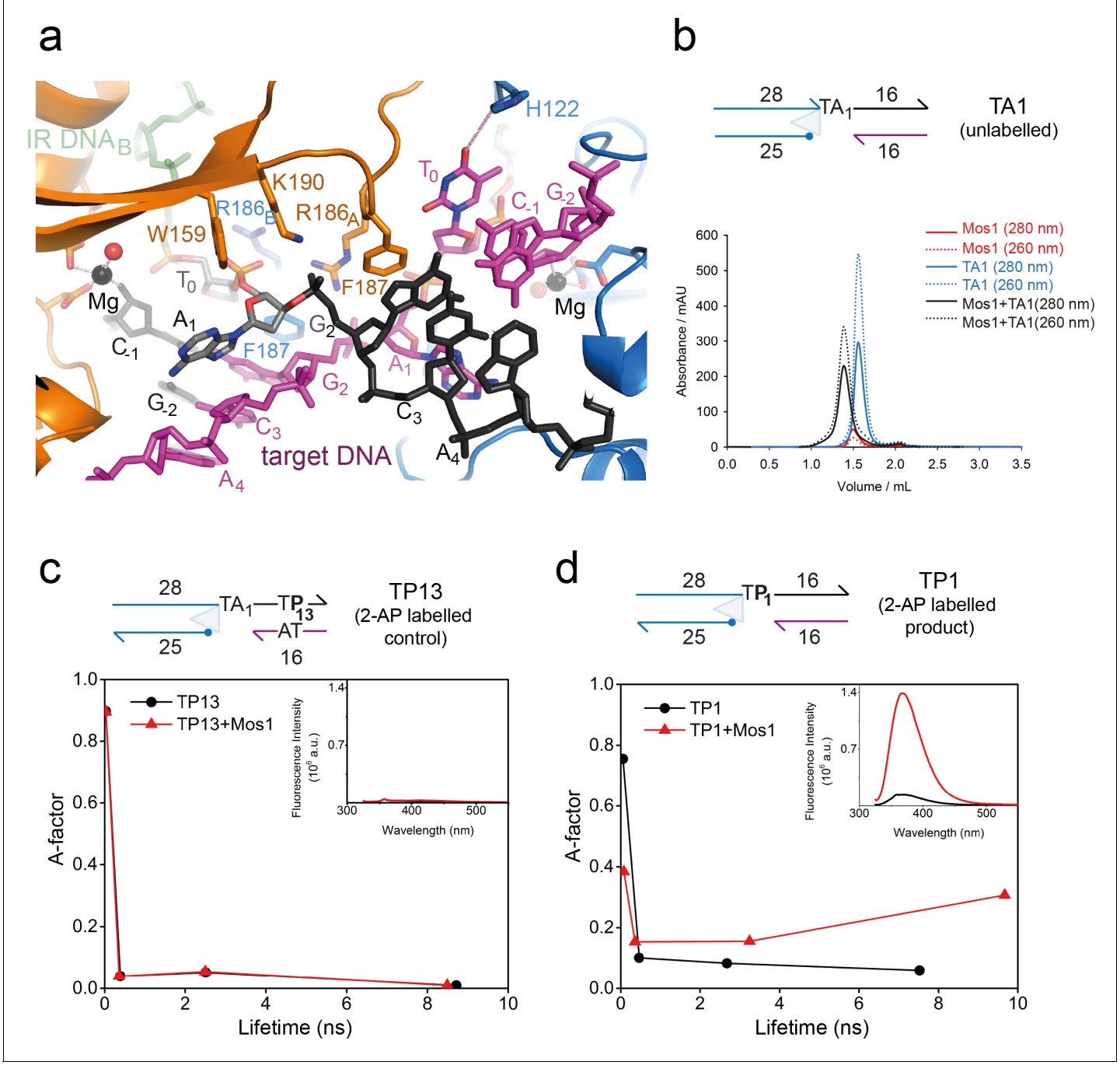

**Figure 3.** Dynamic base flipping of the target adenines. (**a**) Target DNA binding in the Mos1 STC, showing the flipped $A_1$ conformation. The unpaired $T_0$ base stacks with the $C_{-1}$ base of the same strand. See *Figure 3—figure supplement 1* for the effect on strand transfer activity of the mutation H122A. (**b**) Schematic of the TA1 DNA duplex and gel filtration chromatograms of Mos1 transposase (red), TA1 (blue) and the STC (black). UV absorbance at 280 nm (solid line) and 260 nm (dotted line). (**c** and **d**) Fluorescence spectroscopy of the 2AP-labelled DNA oligonucleotides TP13 and TP1, shown schematically in (**c**) and (**d**) respectively. The A-factor (fractional population) and lifetime of each of the four fluorescence decay components are plotted for TP13 and TP1 alone (black circles and lines) and in the presence of Mos1 transposase (red triangles and lines); and tabulated in *Figure 3—source data 1*. The steady-state fluorescence emission spectra are inset in each case.

The following source data and figure supplement are available for figure 3:

**Source data 1.** Fluorescence decay parameters for 2AP-containing duplexes, TP13 and TP1, in the absence and presence of Mos1 transposase.

**Figure supplement 1.** Strand transfer assay comparing the activity of T216A and H122A/T216A Mos1 transposases.

## The target thymines are unpaired and flexible

Each transposon–target thymine junction is clearly defined in the electron density map (*Figure 2d*). There is also clear density for the nucleobase of $T_0$ on strand b (magenta) in active site B. However, we observed no clear density for the $T_0$ base on strand t (active site A), indicating some disorder in its position. Therefore this nucleotide was built as abasic. In active site B the $T_0$ nucleobase π–π stacks with the base of $C_{-1}$ to which $T_0$ would have been connected before strand transfer. The $T_0$ base is unpaired and O4 is 3.4 Å from the H122 imidazole NH, suggesting a possible base-specific hydrogen bond (*Figure 3a*). However, the mutation H122A had no effect on the strand transfer efficiency (*Figure 3—figure supplement 1*), and we conclude that the putative thymine-specific hydrogen bond is not required for target integration and may be transient, due to $T_0$ base mobility.

## The target adenines are flipped into extra-helical positions

The most striking feature of the Mos1 STC structure is flipping of both target adenine ($A_1$) bases of the symmetrical TA sequence into extra-helical positions (*Figure 3a*). The phosphate backbone atoms of $A_1$ and $G_2$ are rotated by ~180°, with respect to the adjacent nucleotides, so that the $A_1$ bases cannot pair with their complementary $T_0$. Instead each unpaired $A_1$ is wedged against the ribose face of the complementary target strand, at an oblique angle to bases $G_2$ and $C_3$ (*Figure 3a*). The aromatic ring of each F187 occupies the space vacated by a flipped $A_1$, forming a π–π stack with the adjacent $G_2$ nucleobase, stabilising this conformation (*Figure 3a*).

To confirm $A_1$ flipping in solution, and to investigate the extent and dynamics of this distortion, we performed fluorescence experiments with DNA containing the adenine analogue 2-aminopurine (2AP). 2AP is an exquisitely sensitive probe of chemical environment: its fluorescence is highly quenched by stacking with the DNA bases and, hence, is sensitive to local duplex structure and enzyme-induced distortion of that structure (*Jones and Neely, 2015*).

We designed three DNA duplexes, each mimicking the strand transfer product: TA1, an unlabelled control analogous to the oligonucleotide used for STC crystallisation (*Figure 3b*); TP13, a labelled control, with 2AP in place of $A_{13}$ on the top target strand (black, *Figure 3c*), where it is base-paired and stacked in duplex DNA; TP1, with 2AP in place of $A_1$ (*Figure 3d*), where the unpaired 2AP is stacked with the adjacent $T_0$ and $G_2$ bases. Upon addition of Mos1 transposase to TA1, we observed complete formation of a nucleoprotein complex by gel-filtration chromatography (*Figure 3b*). When Mos1 transposase was added to TP13, there was no measurable change in the (very low) steady-state fluorescence intensity at 367 nm (inset, *Figure 3c*), consistent with no change in the 2AP environment upon STC formation. In contrast, there was a ten fold increase in fluorescence intensity when Mos1 transposase was added to TP1 (inset, *Figure 3d*), consistent with 2AP at the target site becoming unstacked by flipping into an unquenched, extra-helical environment in the Mos1 STC.

A dynamic picture of DNA conformations in solution can be gained from the interpretation of the fluorescence decay of 2AP, measured by time-resolved fluorescence spectroscopy. In duplex DNA the exponential decay of 2AP fluorescence is typically described by four lifetime components, each reporting on different quenching environments that 2AP experiences as a result of the conformational dynamics of the duplex. The lifetime indicates the degree of quenching (stacking) in a particular conformation and the corresponding A–factor indicates the fractional occupancy of that conformation. The shortest lifetime ($\tau_1 \cong 50$ ps) is due to a highly stacked conformation, which typically accounts for >70% of the population. The longest lifetime ($\tau_4 \cong 9$–10 ns) corresponds to an unstacked conformation in which 2AP is extra-helical and solvent-exposed; this conformation is typically <5% of the population. The intermediate lifetimes ($\tau_2 \cong 500$ ps and $\tau_3 \cong 2$ ns) are due to conformations in which 2AP is intra-helical but imperfectly or partially stacked.

We measured the fluorescence decays of the 2AP-containing DNA duplexes TP13 and TP1 in the absence and presence of Mos1 transposase (*Figure 3c,d*). In the absence of transposase, 90% of the 2AP population of TP13 has the shortest lifetime ($\tau_1=30$ps), indicating a tightly stacked duplex structure. Upon addition of Mos1 transposase, the decay parameters are essentially unchanged showing that the local duplex structure is unaffected, confirming the steady-state fluorescence results. TP1 fluorescence decay, in the absence of protein, is also dominated by the shortest lifetime, stacked component (76%, $\tau_1= 50$ ps, *Figure 3d*), with only 6% of the population in the unstacked state ($\tau_4 = 7.5$ ns). (The differences in the decay parameters between TP13 and TP1 are consistent with a less

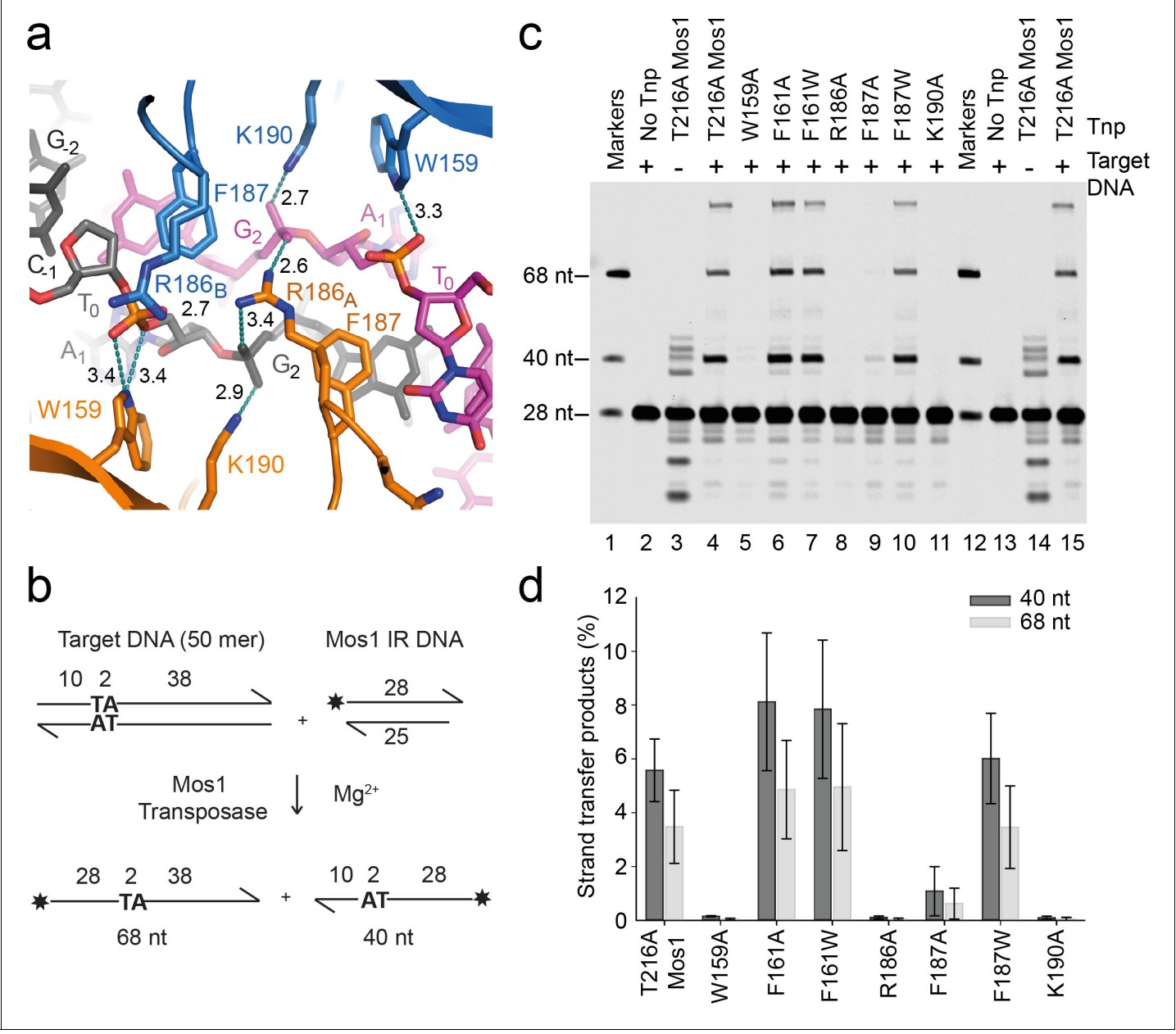

**Figure 4.** Transposase interactions with rotated backbone phosphates stabilise the target DNA. (a) Target DNA phosphate interactions with catalytic domain residues. The side-chains of R186, W159 and K190 can form hydrogen bonds (dotted lines) with backbone phosphate oxygens of $A_1$ and $G_2$ (distances in Å). (b) Schematic of the in vitro Mos1 strand transfer assay. Integration of the 28 nt TS into the top target strand, yields a 68 nt product, whereas integration into the bottom strand gives a 40 nt product. (c) Denaturing PAGE of the strand transfer reaction products. Lanes 1 and 12 contain markers; lanes 2 and 13, reactions without transposase; lanes 3 and 14, reactions without target DNA. (Integration occurs at the two TA dinucleotides in the IR sequence). (d) Quantification of the 40 nt and 68 nt products (as a percentage of total DNA) for each mutant transposase; error bars represent the standard deviation and were calculated from 3 experiments.

tightly stacked environment in the latter, where 2AP is unpaired). However, upon addition of transposase to TP1, the decay parameters change markedly (*Figure 3d*). Most notably, there is a large transfer of population from the highly stacked state ($\tau_1$ = 80 ps) to the unstacked, unquenched state ($\tau_4$ 9.7 ns); the population of the former falls to 38% and that of the latter increases concomitantly to 31%. This clearly confirms that, in solution, 2AP at the position of the target adenine $A_1$ in the Mos1 STC experiences base-flipping into an extra-helical environment. Moreover, flipping of this 2AP is

dynamic: a number of conformational states are sampled, including base-flipped and base-stacked environments.

## Transposase residues stabilise the distorted target DNA backbone

Base flipping of each $A_1$ severely distorts the surrounding target DNA. Side-chain atoms of transposase residues R186, K190 and W159 stabilise these distortions (*Figure 4a*) by forming salt bridges or hydrogen bonds with the $A_1$ and $G_2$ phosphates. The DNA backbone rotations bring the $G_2$ phosphates on both target DNA strands within 6.7 Å of each other and close to the guanidinium group of R186 in subunit A ($R186_A$); each $N\eta H_2$ group hydrogen bonds with a $G_2$ phosphate oxygen on one strand (*Figure 4a*). In both subunits, the K190 side-chain $N\zeta H_2$ forms a salt bridge with the other $G_2$ phosphate oxygen on one strand. Furthermore, the W159 indole $N_1H$ interacts with an $A_1$ phosphate oxygen. Additionally, the $N\eta H_2$ group of $R186_B$ (which has a different conformation to $R186_A$) interacts with the other $A_1$ phosphate oxygen on the top strand (black, *Figure 4a*). The $N\zeta H_2$ of K190 is 5.2 Å from the W159 indole ring and forms a cation-π stack, further enhancing stability. Together these extensive transposase–DNA backbone phosphate interactions stabilise the distorted conformation of the strand transfer product.

## Residues that stabilise the transposition product are required for strand transfer in vitro

Consistent with the structural roles of W159, R186, F187 and K190 in the STC, individual substitution of each of these residues with alanine severely reduced the in vitro strand transfer activity of transposase (*Figure 4b*). We detected <0.03% integration of fluorescently labelled Mos1 IR DNA into a

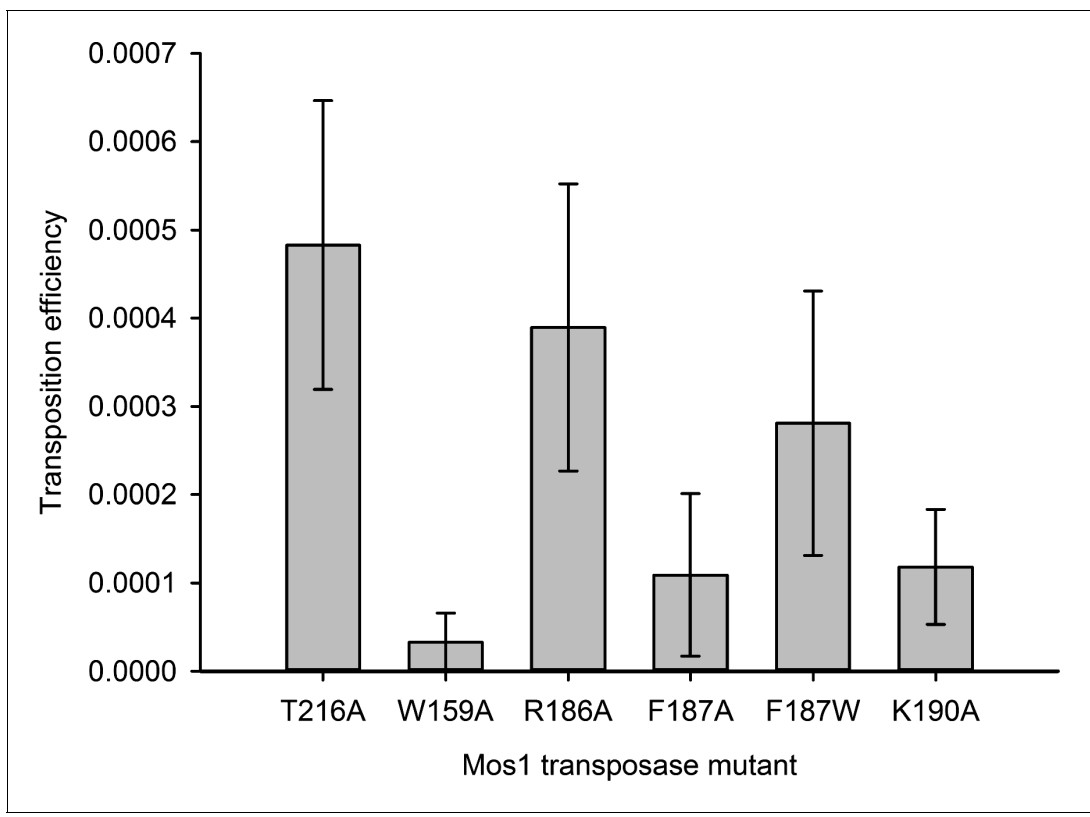

**Figure 5.** Residues that stabilise the transposition product are required for efficient Mos1 transposition in vitro. Efficiencies of an in vitro Mos1 hop assay, performed using Mos1 transposase mutants and donor plasmids containing a kanamycin resistance gene flanked by Mos1 inverted repeats, as described previously (*Trubitsyna et al., 2014*). Excision of the IR-flanked gene from a circular plasmid by transposase, and its integration into a supercoiled target plasmid, results in transfer of the kanamycin resistance to the target plasmid. Each mutant transposase also contained the mutation T216A, which allows soluble protein expression. Sequencing of the transposition products revealed that each mutant transposases retained faithful integration at TA sites. Error bars represent the standard deviation, calculated from three repeats of two experiments.

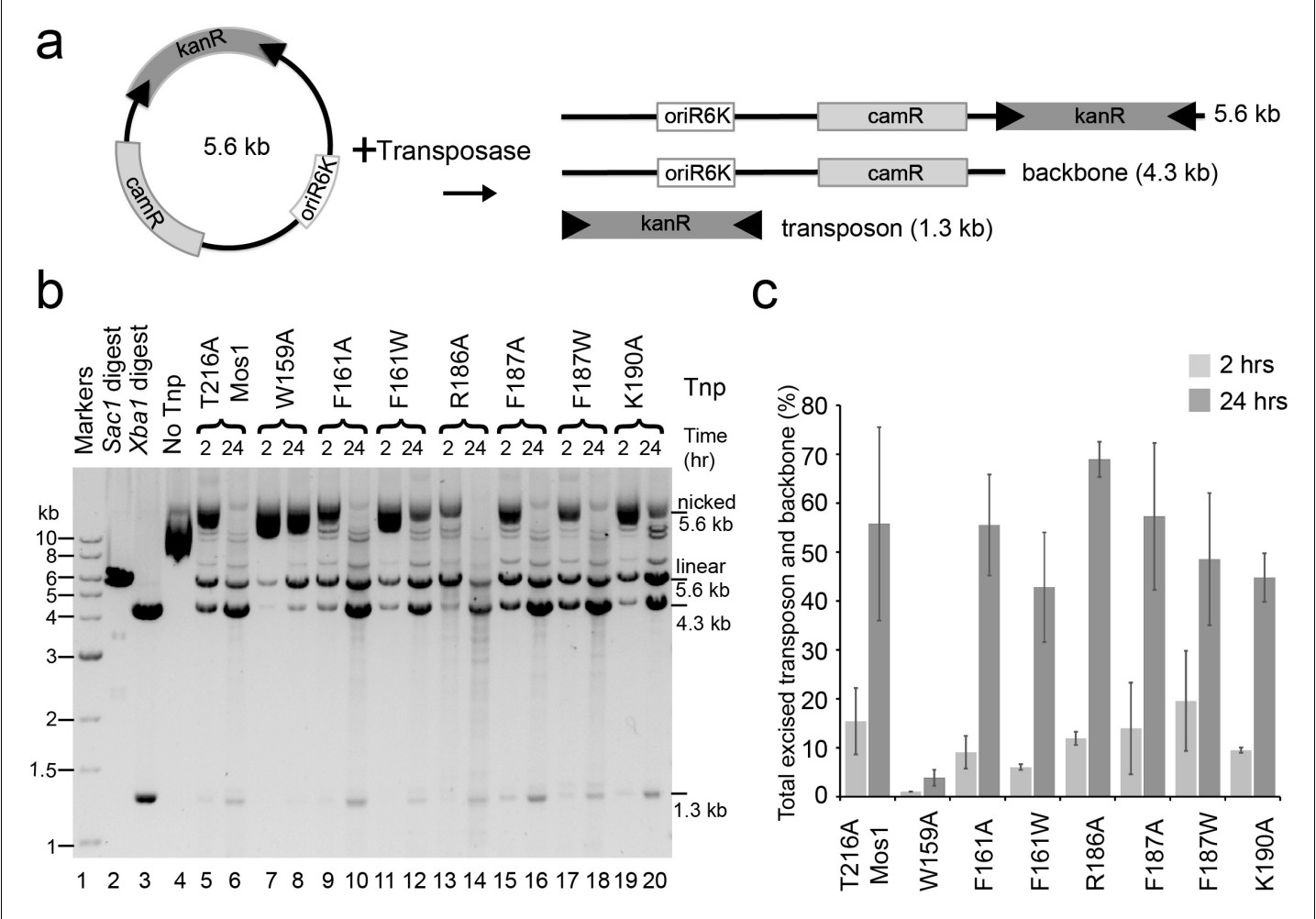

**Figure 6.** Plasmid-based transposon cleavage assays. (a) Schematic of the in vitro plasmid-based Mos1 cleavage assay. (b) Agarose gel showing the products of plasmid-based transposon cleavage assays, for each mutant transposase (Tnp) after 2 hr and 24 hr. Control experiments show linearization of the plasmid with *Sac1* (lane 2), excision of the transposon by *Xba1* digestion (lane 3) and reaction with no transposase (lane 4). (c) Quantification of the transposon and plasmid backbone released (as a percentage of total DNA) after 2 hr and 24 hr. Error bars represent the standard deviation calculated from 2 experiments.

target DNA duplex with a sole TA, using transposases containing the mutation W159A, R186A, K190A or F187A (*Figure 4c,d*). By contrast, the F187W substitution resulted in 9.5% strand transfer, compared to 9.1% with T216A Mos1 transposase. Thus, an indole ring, like a phenyl ring, can occupy the space vacated by the flipped $A_1$ base and stabilise the strand transfer product by stacking with the $G_2$ base. The individual substitutions W159A, K190A or F187A also reduced the in vitro transposition efficiency to <20% that of T216A Mos1 transposase (*Figure 5*).

To test if W159, R186, F187 and K190 are also required for transposon excision, we performed a plasmid-based transposon cleavage assay (*Figure 6a*). Transposon excision, and concomitant plasmid backbone release, was not affected by the transposase mutations R186A, F187A, F187W, K190A, F161A or F161W (*Figure 6b,c*). However, the W159A mutant transposase excised only 3.9% of the plasmid after 24 hr, compared to 55.6% for the T216A transposase. Thus, Mos1 transposase residues F187, R186 and K190 are required for target DNA integration, but are not essential for earlier cleavage steps, whereas W159 is required for both excision and strand transfer.

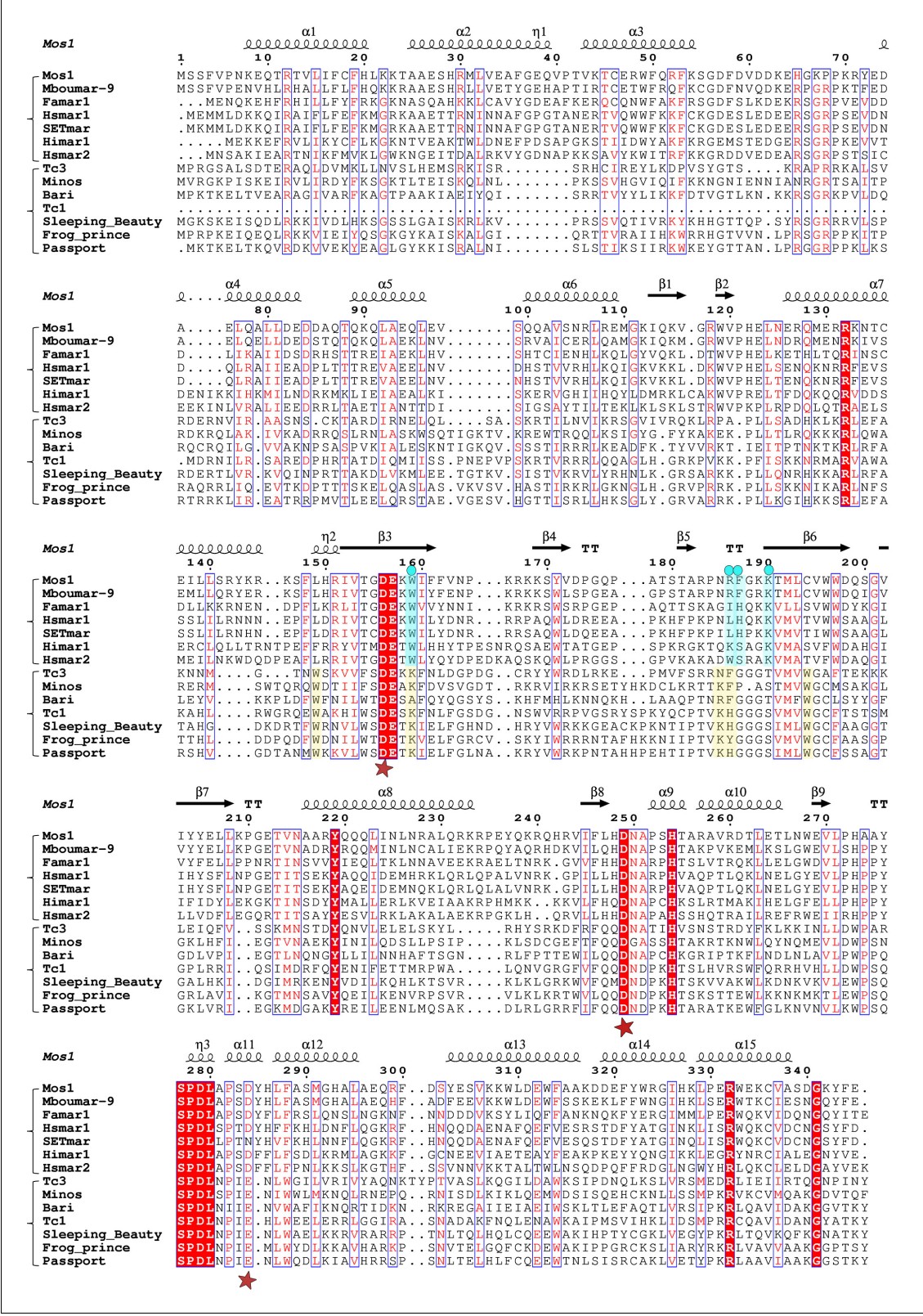

**Figure 7.** Alignment of the amino acid sequence of Mos1 with six other mariner transposases and five Tc1-family transposases. The secondary structure elements of Mos1 transposase in the Mos1 STC are shown above the alignment. A red star below the alignment denotes the position of each of the catalytic acidic residues of the DDE/D triad. The third residue of this triad is typically D in the mariner sub-family and E in the Tc sub-family. The key

*Figure 7 continued on next page*

*Figure 7 continued*

residues involved in target DNA stabilisation in the Mos1 STC are highlighted in blue and marked by a blue dot. The figure was created with ESPript 3.0 (http://espript.ibcp.fr/ESPript/cgi-bin/ESPript.cgi).

## Target DNA stabilising amino acids are conserved in mariner transposases

Alignment of the Mos1 transposase sequence, with other mariner and Tc1-like transposases (*Figure 7*), reveals that K190 and W159, which form a cation-π stack and interact with target DNA phosphates, are strictly conserved among mariner transposases. Despite the crucial role of Mos1 R186 for strand transfer in vitro, this residue is not conserved in all mariner transposases. However, the aromatic nature of F187 is conserved as either F or H in most other mariner transposases. Thus, many of the target-stabilising interactions observed in the Mos1 STC may also exist in other mariner transposases.

The Tc1-like sequences have a conserved lysine at the position equivalent to W159 in Mos1 and there are two, proximal conserved tryptophans – aligned with Mos1 residues 149 and 194 – which could fulfil the role of W159 in Mos1 (*Figure 7*). Furthermore, the Tc1-like transposases contain either K or R one amino acid upstream of R186 in Mos1, followed by an aromatic residue: F, H or Y. These residue pairs could stabilise target DNA in a similar way to R186 and F187 in Mos1. Thus, there may be common features in the target DNA integration mechanisms of the two branches of the *mariner/Tc1* family.

## Transposase recognises the flipped target adenines via base-specific interactions

The target-stabilising interactions described above are non-specific. In contrast, in the flipped conformation, the Watson-Crick face of each unpaired $A_1$ base makes two adenine-specific hydrogen bonds with V214 backbone atoms (*Figure 8a*): the exocyclic 6-amine interacts with the carbonyl oxygen, and N1 interacts with the backbone amide.

To test if these adenine-specific hydrogen bonds are important for transposon integration specifically at a TA, we performed in vitro strand transfer assays with 2AP-containing target DNA. The arrangement of H-bond donors differs between adenine and 2AP (*Figure 8b*). Therefore, by replacing each $A_1$ with 2AP we expect to lose the H-bond between the $A_1$ 6-amino and V214 CO, and introduce a steric clash between the 2-amino of 2AP and T213 $C_\alpha$. Since 2AP, like adenine, can make two hydrogen bonds in a base pair with thymine (*Figure 8b*), replacing $A_1$ with 2AP is unlikely to alter the stability and bendability of duplex target DNA.

We found that replacing both $A_1$s with 2AP (*Figure 8c,d*) resulted in a dramatic loss of specific integration at each $T_0$ 5' of the 2AP, consistent with the predicted loss of adenine-specific hydrogen-bonds with transposase. Our fluorescence experiments show that 2AP at position 1 in target DNA undergoes dynamic base flipping in the Mos1 STC (*Figure 3d*), whereas our crystallographic snapshot with adenine at the equivalent position suggests a static flipped conformation. This may reflect different experimental conditions: the fluorescence experiments were performed in solution at room temperature, whereas the crystal structure was obtained at cryogenic temperatures. However, it is also consistent with a lack of specific interactions between 2AP and transposase, leading to an inability to trap the flipped 2AP conformation. We conclude that Mos1 integration at TA requires adenine-specific interactions with transposase to trap the flipped $A_1$ conformation.

Next we asked which $A_1$ of the symmetrical TA sequence is essential for integration at $T_0$: the adjacent $A_1$ on the same strand or the complementary $A_1$. We replaced each $A_1$ individually with 2AP, and efficient Mos1 integration occurred at a $T_0$–$A_1$ step when the $T_0$ was base-paired with 2AP, but was reduced at a $T_0$–$2AP_1$ step (*Figure 8c,d*). We conclude that specific Mos1 integration at a $T_0$ requires trapping of the flipped $A_1$ adjacent to it on the same strand.

Finally we asked if the lower stability of a T:A base-pair, compared to G:C, favours *mariner/Tc1* transposon integration at TA sites. We predicted that $A_1$ flipping, and therefore strand transfer, would be hindered if the T:A base-pairing was strengthened by a third hydrogen bond, but enhanced with weakened base pairs. We replaced both $A_1$s with 2,6-diaminopurine (2-amino-dA, or D), which forms three hydrogen bonds with dT (thereby increasing base-pair stability) but only one

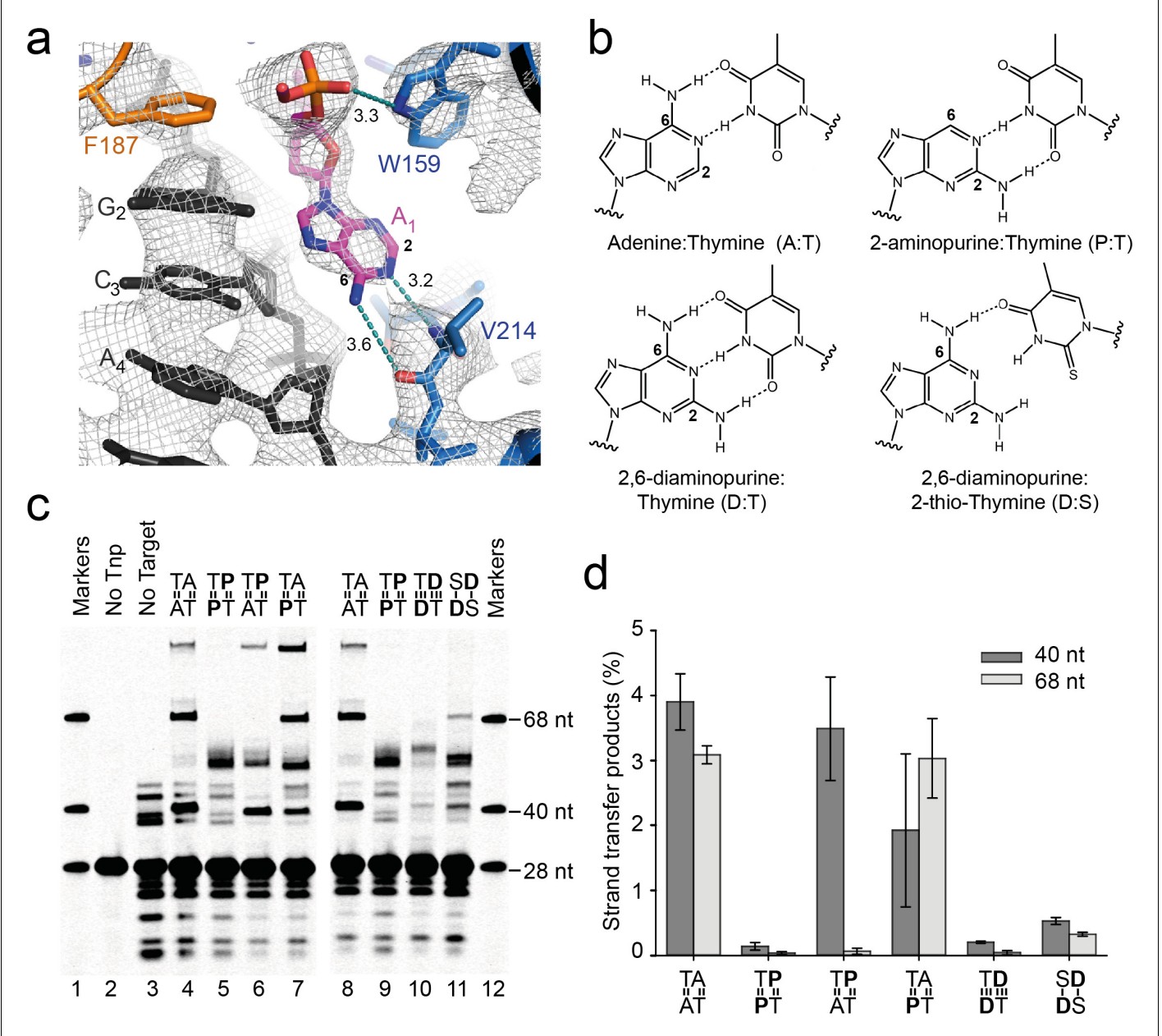

**Figure 8.** Base-specific recognition of the flipped adenine. (a) Close up view of one of the flipped target adenines in the Mos1 STC crystal structure showing the hydrogen bond interactions (dotted cyan lines, distance in Å) with the V214 backbone atoms and the 2 and 6 positions of the adenine ring. The simulated annealing composite omit 2Fo-Fc electron density map (grey mesh) is contoured at 1.2σ. (b) Chemical structures and base-pairing of adenine, A, and its analogues 2-aminopurine, P, and 2,6-diaminopurine, D, with thymine, T or 2-thio-thymine, S. A steric clash between the 2-thio group of S and the 2-amino group of D tilts the bases relative to each other, and thus only one H-bond forms. (c) Denaturing PAGE of the products of strand transfer reactions with target DNA containing adenine and/or thymine analogues, as indicated above lanes 4 to 11. (d) Quantification of the 40 nt and 68 nt strand transfer products for each target DNA duplex, as a percentage of total DNA. Error bars represent the standard deviation, calculated from 2 experiments.

hydrogen bond when paired with 2-thio-dT (or S) (*Kutyavin et al., 1996*) (*Figure 8b*). 2-amino-dA can form the adenine-specific interactions with V214 seen in the Mos1 STC structure, however the 2-amino group adds a potential clash with transposase that could lead to reduced specificity.

We compared Mos1 strand transfer into TA and the altered target sequences TD and SD, with strengthened and weakened base pairing respectively (*Figure 8c*, lanes 10 and 11). We measured

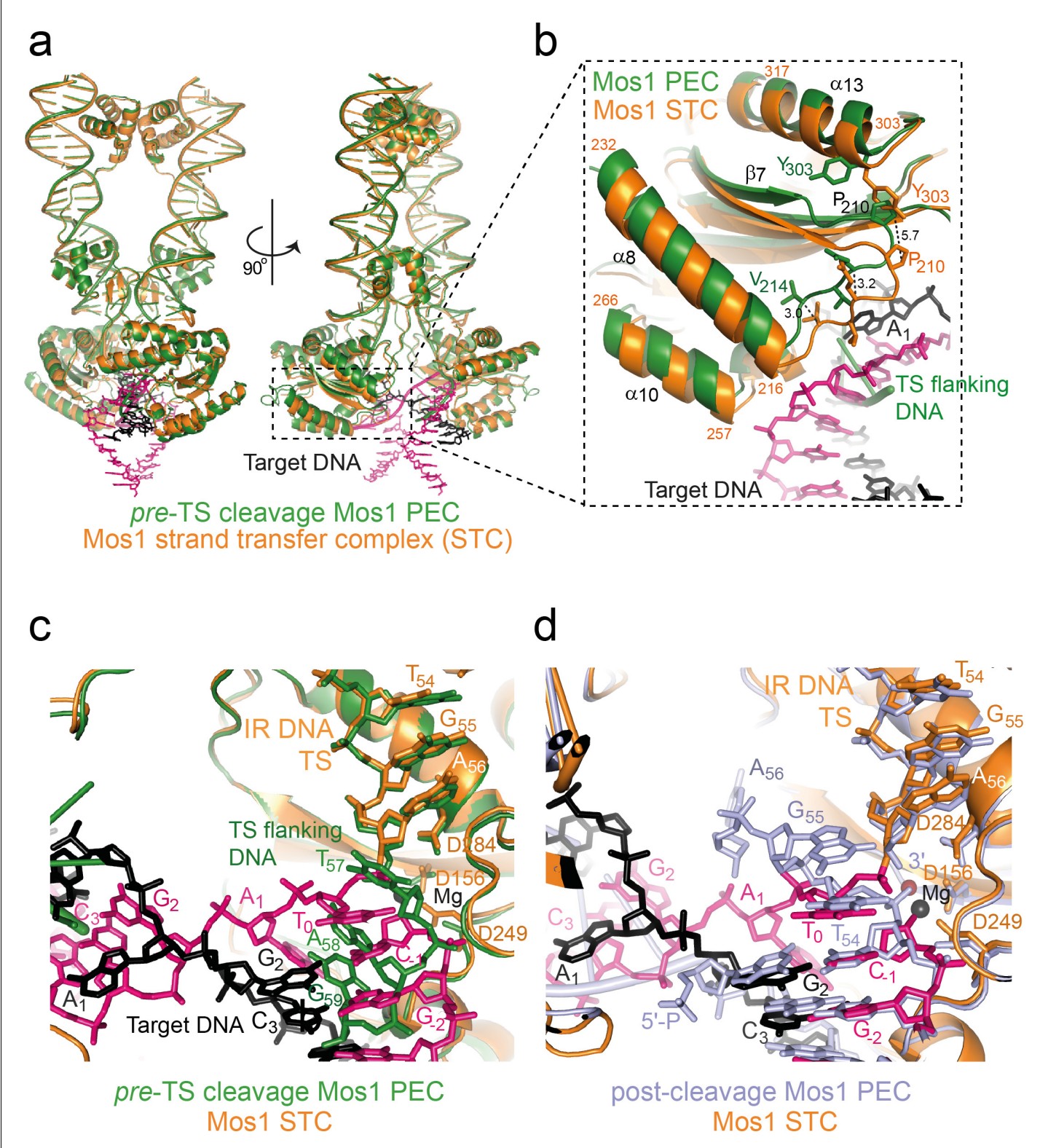

**Figure 9.** Structural comparison of the Mos1 STC with the pre- and post-TS cleavage Mos1 paired-end complexes. (**a**) Orthogonal views of the Mos1 STC (orange) superimposed on the *pre*-TS cleavage PEC (PDB ID: 4U7B, green): r.m.s.d. over all transposase backbone atoms, 1.2 Å. *Video 1* and *video 2* show the transposase morphing from the *pre*- and *post*-cleavage PEC structures to the STC, respectively. (**b**) Close-up view of part of the catalytic domain, boxed in (**a**). Mos1 STC target DNA and the *pre*-TS cleavage PEC flanking DNA are shown as sticks (pink and black) and a green cartoon, respectively. Dotted lines indicate the displacement between the two structures, with distances in Å. (**c**) and (**d**) Close-up view of the Mos1 STC

*Figure 9 continued on next page*

*Figure 9 continued*

(orange) active site superimposed on (**c**) the *pre*-TS cleavage PEC (green) and (**d**) the *post*-TS cleavage PEC (PDB ID: 3HOS): $T_{54}$ in the additional DNA duplex (lavender sticks) may represent $T_0$ of target DNA before strand transfer. A full view of the Mos1 STC superposed on the *post*-TS cleavage PEC structures is shown in ***Figure 9—figure supplement 1***.

The following figure supplement is available for figure 9:

**Figure supplement 1.** Structural comparison of the Mos1 STC with the post-TS cleavage Mos1 paired-end complex.

6.99% integration into TA, but only 0.24% integration into the TD sequence (***Figure 8c,d***) and 0.86% integration into SD (***Figure 8c***); in the latter experiment many other, non-specific integration products were also observed. Thus, the weakness of the T:A base pair promotes integration at the TA sequence, and the pattern of H-bond donors and acceptors on the Watson-Crick face of adenine is important for specificity.

## Discussion

The Mos1 STC structure provides a snapshot of Mos1 transposition in the post-integration state. The severe target DNA bend (~147°) is consistent with a bias for *mariner/Tc1* integration at highly bendable, palindromic AT-rich sequences (***Vigdal et al., 2002***; ***Yant et al., 2005***). Studies by Pflieger *et al.* suggested that target DNA also bends before Mos1 strand transfer (***Pflieger et al., 2014***). Comparison of the Mos1 STC structure with our previous TCC model (containing straight target DNA) and both the *pre*- and *post*-TS cleavage PECs (***Dornan et al., 2015***; ***Richardson et al., 2009***) supports this conclusion. Our previous TCC model (***Richardson et al., 2009***) of straight B-form target DNA binding highlighted clashes with some transposase loop residues, indicating conformational changes in the target DNA and/or the transposase would be required for target capture. The similar architectures and interactions of the IR DNA and transposase in the STC and both PEC structures (***Figure 9a*** and ***Figure 9—figure supplement 1***) suggest that target DNA is likely deformed. Changes to the transposase conformation are subtle and include closing-in of the catalytic domain towards the target DNA after strand transfer (***Videos 1*** and ***2***). The largest displace-

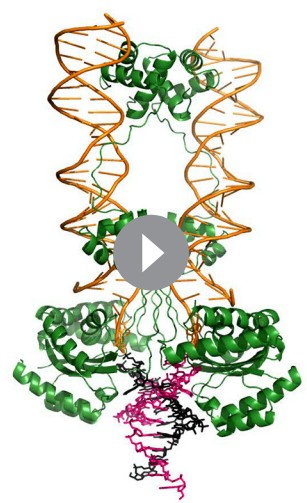

**Video 1.** Morphing of the Mos1 transposase conformation in the *pre*-TS cleavage PEC (PDB ID: 4U7B) into the Mos1 STC conformation. Related to ***Figure 9***.

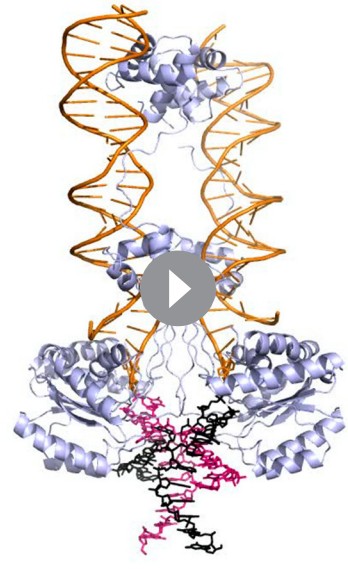

**Video 2.** Morphing of the Mos1 transposase conformation in the *post*-cleavage PEC (PDB ID: 3HOS) into the Mos1 STC conformation. Related to ***Figure 9***.

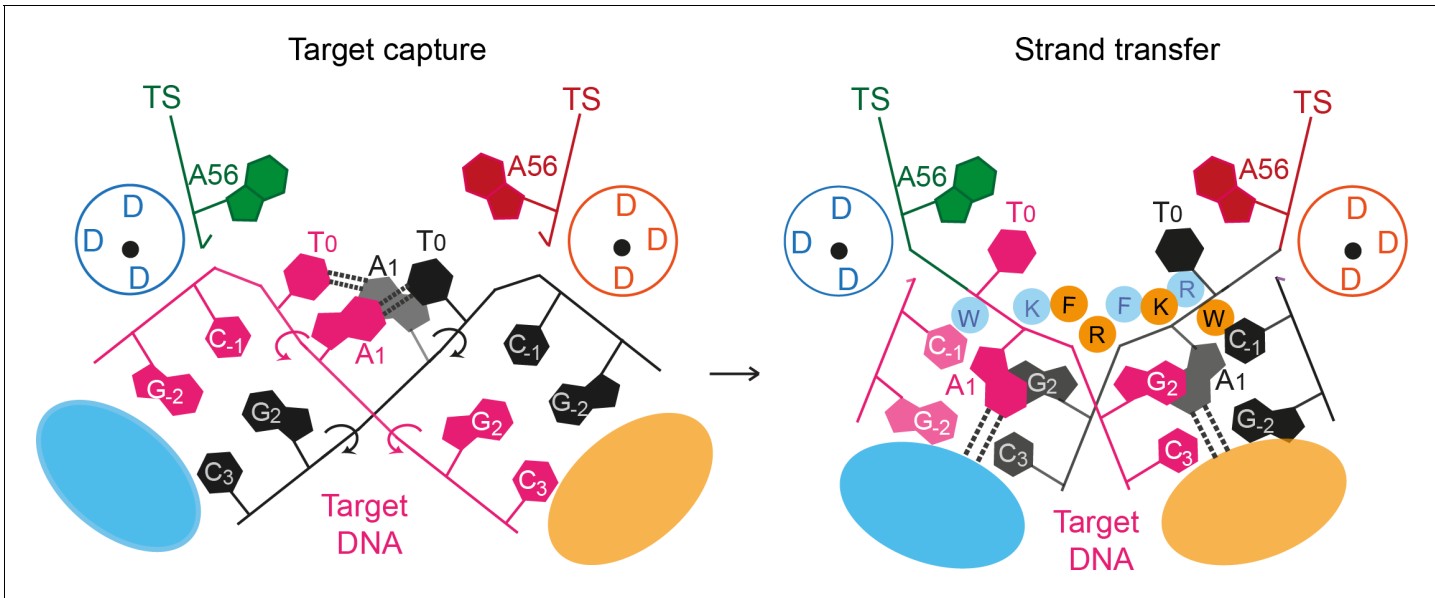

**Figure 10.** A proposed mechanism for Mos1 transposon integration incorporates target DNA bending and trapping of flipped target adenines. Schematic representation of key features of the STC (right) and the proposed target capture complex (left), with transposase subunits (orange and blue). Filled circles represent residues W159 (W), R186 (R), F187 (F) and K190 (K) and the encircled DDD depicts each active site. Arrows indicate rotation of the DNA backbone of each target DNA strand (magenta and black). Dotted lines represent hydrogen bonds between TA base pairs in the TCC and between flipped target adenines and transposase backbone atoms in the STC.

ment (5.7 Å) is at P210 in the turn between $\beta$7 and $\alpha$8 and around helices $\alpha$8 and $\alpha$10, which cradle the target DNA (*Figure 9b*). $T_0$ in the Mos1 STC is in a different orientation to the thymine ($T_{57}$) of the flanking target site duplication in the *pre*-TS cleavage PEC (*Figure 9c*), which is recognised by base-specific interactions with the WVPHEL motif (*Dornan et al., 2015*). By contrast, $T_0$ closely aligns with $T_{54}$ of the additional DNA duplex in the *post*-TS cleavage PEC (*Figure 9d*), which may represent the target strand before integration.

Strain created by target DNA bending during target capture likely drives the phosphate backbone rotations that flip the target adenines into extra helical positions (*Figure 10*). Subsequent trapping of the flipped adenines may correctly position the scissile target phosphates for in-line attack by the cleaved transposon ends. Breaking of the target DNA strands by strand transfer would allow displacement of the new transposon-target DNA junction from the active site, preventing reversal of the reaction. Structural and biochemical characterisation of the target capture complex will illuminate this sequence of events.

Many DNA-metabolising enzymes use base flipping to expose bases normally embedded within a double helix; enabling base methylation (*Klimasauskas et al., 1994*), removal of damaged or mismatched bases or for DNA sequence recognition (*Bochtler et al., 2006*; *Neely et al., 2009*). During Tn5 transposon excision, formation and resolution of DNA hairpins at the transposon ends requires base flipping: rotation of a base close to the cleavage site, into a protein pocket, relieves strain in the tight hairpin bend and aligns the transposon ends for cleavage (*Ason and Reznikoff, 2002*). Similarly, DNA hairpin stabilisation by base flipping has been proposed for V(D)J recombination and transposition of Hermes and Tn10 (*Lu et al., 2006*; *Bischerour and Chalmers, 2009*).

Active intrusion of a probe amino acid residue can drive base flipping. During Mos1 integration, the F187 aromatic ring may actively force the $A_1$s from the DNA helix, similar to the methionine probe in Tn10 transposon excision (*Bischerour and Chalmers, 2009*). Alternatively, F187 may passively fill the gap left by $A_1$ after flipping, to enhance the stability of the distorted target DNA conformation. In this scenario, the conserved transposase residues K190 and W159 may be alternative drivers of the base-flipping rotations.

Target adenine–specific interactions with V214 backbone atoms suggest a molecular basis for TA target sequence recognition in the post-integration state. The structured loop containing V214 has

the consensus sequence T-(V/I)-(N/T) in mariner transposases (*Figure 7*), suggesting that the role of transposase backbone atoms in TA recognition may be conserved among mariner-family transposases. The Tc1-family transposases, including Sleeping Beauty, also display sequence conservation in this region (*Figure 7*), suggesting similar recognition mechanisms exist in that closely-related family. Structures of Sleeping Beauty transposition intermediates will reveal if this is the case.

Target DNA bending is a recurring theme in DNA transposition by DDE/D recombinases. The severe target DNA bend (~147°) observed in our Mos1 STC structure is similar to the ~140° target DNA distortion in the bacterial MuA transpososome, which was proposed to drive the isoenergetic strand transfer reaction forward (*Montaño et al., 2012*). Mu employs a helper protein (MuB) in its transposition, which may facilitate target DNA bending by forming helical filaments on DNA, prior to capture by the MuA transpososome (*Mizuno et al., 2013*). Similarly, the bacterial insertion sequence IS*21* requires IstB for efficient transposition. In the presence of ATP, IstB self-assembles into decamers that can bend ~50 bp DNA by 180° (*Arias-Palomo and Berger, 2015*). In the PFV integrase target capture and strand transfer complexes (*Maertens et al., 2010*) naked target DNA is bent by 55°. Nucleosomal DNA is peeled from the histone octamer and similarly deformed by interactions with the PFV intasome, providing a structural basis for retroviral integration at nucleosomes (*Maskell et al., 2015*). By contrast, recent evidence indicates that *mariner/Tc1* transposons preferentially integrate at linker regions between nucleosomes (*Gogol-Döring et al., 2016*). Our results provide a structural basis for this preference: severe target DNA bending (~147°) by the transpososome can be more easily achieved on flexible linker DNA than on DNA tightly engaged with the histone octamer in a nucleosome structure.

Taken together our structural and biochemical data support a dynamic bend, flip and trap mechanism for *Mos1* transposon integration at TA target sites (*Figure 10*) that may be a conserved feature of *mariner/Tc1* transposition. As such, our results provide a framework for designing mariner/Tc1 transposases with modified target specificities.

# Materials and methods

## Transposase mutation, expression and purification

Expression constructs encoding Mos1 transposase mutants H122A, W159A, F161A, F161W, R186A, F187A, F187W, K190A were generated by site-directed mutagenesis (Quikchange, Stratagene) of the codon-optimised Mos1 gene (*Trubitsyna et al., 2014*), according to the manufacturer's protocol. Each plasmid also incorporated the T216A mutation allowing soluble expression of Mos1 transposase in *E. coli* (*Richardson et al., 2004*). Each mutant transposase was expressed and purified as described previously (*Richardson et al., 2004*), exchanged into buffer containing 25 mM PIPES pH 7.5, 250 mM NaCl, 0.5 mM DTT and 50% (v/v) glycerol and concentrated to between 10–20 mg mL$^{-1}$.

## Preparation of ds DNA substrates

The sequences of all DNA oligonucleotides are shown in *Table 1*. HPLC purified oligonucleotides for crystallisation of the STC were purchased from IDT (Belgium), PAGE purified and dissolved to 1 mM in TEN buffer (10 mM Tris pH 8, 1 mM EDTA, 50 mM NaCl). The 36 nt TS incorporates the 28 nt IR and target DNA (as shown in *Figure 1c*). The 25 nt NTS is complementary to the TS IR DNA sequence and represents the authentic product the first cleavage. The 10 nt target DNA sequence, includes six nucleotides complementary to the 3' TS target sequence and four self-complementary nucleotides (cohesive 5' ends). The three oligonucleotides were mixed in a 1:1:1 molar ratio and annealed by heating to 363 K for 3 min and cooling to room temperature over ~2 hr.

For time-resolved fluorescence experiments, DNA oligonucleotides were synthesised and HPLC purified by ATDBio (Southampton, UK). Three TS sequences, extended at the 3' end to 46 nt, were synthesised: TS_A1, an unlabelled control; TS_P1, with 2AP in place of the target adenine; TS_P13 with 2AP at position 13, another control. Each TS was annealed with the 25 nt NTS and the 16 nt target_16 sequence complementary to the TS 3' end. This yielded the three duplexes – TA1, TP1 and TP13 – which mimic the Mos1 strand transfer product.

For the strand transfer assays, the IR DNA was prepared by annealing the 28 nt 5'-IRDye 700 labelled TS with the 25 nt complementary NTS. The 50-mer TA target DNA, was prepared by annealing complementary top and bottom strands (*Table 1*). Five target DNA variants were similarly

prepared: three had 2-aminopurine (P) in place of the target adenine on the top and/or bottom strand. A fourth had 2,6-diaminopurine (D) in place of the target adenine on both strands, and the fifth also had 2-thio-thymine (S) in place of the target thymine on both strands. The annealed IR and target oligonucleotides were purified by HPLC.

## Preparation of the Mos1 STC

The STC was formed by adding T216A Mos1 transposase (438 µM) and STC ds DNA (229 µM) together to final concentrations of 50 µM each in a solution of 25 mM PIPES-NaOH pH 7.5, 250 mM NaCl, 20 mM $MgCl_2$ and 1 mM DTT. The final concentration of the STC was 25 µM.

## Crystallisation

Crystals were grown by sitting drop vapour-diffusion. Drops contained 2 µL of STC (25 µM) and 1 µL of well solution comprising 30% (v/v) MPD, 0.1 M sodium cacodylate pH 6.5 and 0.2 M magnesium acetate tetrahydrate. The crystals were cooled in liquid nitrogen for X-ray diffraction experiments.

## X-ray crystal structure determination and refinement

X-ray diffraction data were collected on beam line I02 at the Diamond Light Source. Crystals displayed C-centred (*C*121) symmetry and diffracted X-rays to a maximum resolution of 3.3 Å. The X-ray diffraction data were processed with iMosflm, scaled and merged with AIMLESS and the statistics are shown in *Table 2*. Initial phases were determined by molecular replacement, using our structure of the Mos1 PEC (PDB ID: 3HOS, chains A to F, comprising the transposase dimer and two cleaved IR DNA molecules) as the search model in PHASER. The difference electron density after molecular replacement is shown in *Figure 2—figure supplement 1*. The remaining structure was built manually. Restrained refinement was performed with Refmac and Coot and included automatic non-crystallographic symmetry restraints on the protein and DNA chains. The refinement statistics are shown in *Table 2*. All structural diagrams were prepared using PyMOL (http://www.pymol.org/) and Adobe Illustrator.

## In vitro strand transfer and transposon cleavage assays

Target integration assays were performed as described previously (*Wolkowicz et al., 2014*). 20 µL reactions containing 15 nM of a 50-mer target DNA, 1.5 nM IR DNA and 15 nM Mos1 transposase in buffer containing 25 mM HEPES pH 7.5, 50 mM Potassium Acetate, 10% (v/v) glycerol, 0.25 mM EDTA, 1mM DTT, 10 mM $MgCl_2$, 50 µg/mL BSA and 20% (v/v) DMSO were incubated for two hours at 30°C and the products separated on an 8% denaturing polyacrylamide gel. To visualise the products, the IRDye700 was excited at 680 nm and detected on a LI-COR Odyssey system. The fluorescence intensities of the product bands were quantified using Image Studio software. Plasmid-based transposon cleavage assays were performed as described previously (*Trubitsyna et al., 2014*).

## Steady state fluorescence

Measurements were acquired, in photon counting mode, on a Fluoromax–3 spectrofluorimeter (Jobin Yvon, Stanmore, UK), on samples of the 2AP-containing duplexes TP13 or TP1 (10 µM), alone or mixed with 11 µM Mos1 transposase, in buffer composed of 25 mM PIPES-NaOH pH 7.5, 250 mM NaCl, 20 mM $CaCl_2$, 1 mM DTT. A circulating water bath maintained sample temperatures at 25°C. Emission spectra were recorded in the range 325–550 nm, with an excitation wavelength of 317 nm and excitation and emission bandwidths of 2.5 nm.

## Time-resolved fluorescence

Measurements were performed using time-correlated single photon counting, on an Edinburgh Instruments spectrometer equipped with TCC900 photon counting electronics, as described previously (*Neely et al., 2005*). The excitation source was the third harmonic of the pulse-picked output of a Ti-sapphire femtosecond laser system (Coherent, 10 W Verdi and Mira Ti-Sapphire), consisting of ~200 fs pulses at a repetition rate of 4.75 MHz and a wavelength of 317 nm. The instrument response of the system was ~80 ps full-width at half-maximum.

Fluorescence decay curves were analysed by iterative re-convolution, assuming a multi-exponential decay function, given in *Equation (1)*

$$I(t) = \sum_{i=1}^{4} A_i exp\left(\frac{-t}{\tau_i}\right) \qquad (1)$$

where $I$ is the fluorescence intensity as a function of time (t); $\tau_i$ is the fluorescence lifetime of the $i^{th}$ decay component and $A_i$ is the fractional amplitude (A-factor) of that component.

Decays were collected at two emission wavelengths (375 nm and 390 nm) and were analysed globally, with $\tau_i$ as the common parameter, using Edinburgh Instruments software FAST.

## Acknowledgements

We thank the staff at Diamond Light Source (I02). D Finnegan and A Cook provided insightful comments on the manuscript. The research data supporting this publication can be accessed at 10.7488/ds/1404.

## Additional information

### Funding

| Funder | Grant reference number | Author |
|---|---|---|
| Biotechnology and Biological Sciences Research Council | BJ000884 | Julia M Richardson |
| Wellcome Trust | 085176/Z/08/Z | Julia M Richardson |

The funders had no role in study design, data collection and interpretation, or the decision to submit the work for publication.

### Author contributions

ERM, Performed investigations, Analysed data, Drafted the paper; HG, GM, Performed experiments and analysed results; ACJ, Conceived the fluorescence experiments, Analysed data, Edited the paper; JMR, Conceived the study and the methodology, Analysed X-ray diffraction data and drafted, Reviewed and edited the paper.

### Author ORCIDs

Elizabeth R Morris, http://orcid.org/0000-0003-1893-7515
Julia M Richardson, http://orcid.org/0000-0002-1547-3009

## Additional files

### Major datasets

The following datasets were generated:

| Author(s) | Year | Dataset title | Dataset URL | Database, license, and accessibility information |
|---|---|---|---|---|
| Richardson JM, Morris ER | 2016 | Crystal structure of the Mos1 Strand Transfer Complex | http://www.rcsb.org/pdb/search/structid-Search.do?structureId=5HOO | Publicly available at the RCSB Protein Data Bank (accession no: 5HOO) |
| Elizabeth M, Heather G, Grant M, Anita J, Julia R | 2016 | A bend, flip and trap mechanism for transposon integration | http://dx.doi.org/10.7488/ds/1404 | Publicly available at the The University of Edinburgh DataShare |

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
