## [Decision Letter]

Thank you for submitting your article "A bend, flip and trap mechanism for transposon integration" for consideration by *eLife*. Your article has been favorably evaluated by John Kuriyan as the Senior editor and three reviewers, one of whom is a member of our Board of Reviewing Editors.

The reviewers have discussed the reviews with one another and the Reviewing Editor has drafted this decision to help you prepare a revised submission.

Summary:

By using a combination of structural biology, time-resolved fluorescence experiments and transposition assays, Richardson and colleagues reveal new mechanistic insight into the DNA integration mechanism of the Mos1 transposon. Transposable elements make up a large proportion of most genomes, and their movement from place to place generates genome diversity and contributes to genome instability. This work addresses how Mos1 insertion into new sites in DNA is faithfully achieved. A crystal structure of Mos1 transposase complexed to transposon ends covalently joined to target DNA reveals severe distortion of target DNA and flipping of the target adenines into extra-helical positions. The molecular recognition by transposase of these flipped residues provides a molecular basis for target site selection. Transposase residues that stabilise the DNA distortion are required for efficient transposition.

Essential revisions:

Below the comments that you need to take into account in your revision are listed. Also note that *eLife* expects papers to refer to the organism system in either the title or Abstract so I would propose you add a clause in the Abstract stating briefly that Mos1 is derived from *Drosophila mauritiana*. I should add that all reviewers were enthusiastic about this work, and their view is encapsulated by the statement of one reviewer: 'This paper by Dr. Richardson and co-workers is impressive overall, as it provides novel and substantial insights to Mos1 and target DNA interactions. The experimental strategy to obtain crystals is clever and clearly worked out very nicely. I found the biochemical work, including the fluorescence spectroscopy, persuasive. When taken together, the correlation of the structural results with the solution biochemical data brings this work to a very high standard'.

1) It would help to show how similar the STC is to the previously published transpososome (PEC) structures. A superposition (and/or rmsd values) would be very useful.

Importantly, the molecular replacement was solved with the PEC complex, 3HOS, as the search model; one reviewer recalls that 3HOS has a peculiar packing that was proposed to mimic target DNA. Whether this proposal is still valid or not, the reviewer assumes that the TS strand must have been built in with some kind of difference electron density after molecular replacement found the orientation and the position of the complex. This difference electron density (before any model building) should be shown (at the very least as a supplementary figure) as it is crucial for everything in the paper-this should be explicitly shown. Furthermore, as the resolution is moderate, possible model bias is a concern. The reviewer believes that the electron densities shown are inappropriate, as they are the 2Fo-Fc kind, and therefore subject of the modeling assumptions. In the reviewer's opinion, all the electron density figures should be replaced with simulated annealed omit maps (easily and rapidly computed with the freely available Phenix suite of programs). Considering the similarities of the STC to the previously published PEC structures it is hard to imagine that target DNA would bend *after* strand transfer. Would a straight target DNA fit between active sites of the Mos1 transpososome?

2) Discussion, seventh paragraph: Contrary to the oft-repeated dogma, the ability/propensity of retroviruses to integrate into nucleosomes is not due to the "pre-bent" nature of nucleosomal DNA (which is relatively smooth), but more likely due to complex interactions of the intasome with both gyres of nucleosomal DNA and the histone octamer. The interactions collectively offset the energetic penalty associated with the much more severe bending of the nucleosomal DNA between active sites of the intasome. Related: the second paragraph of the Introduction seems to suggest that nucleosomal DNA is not B-form.

3) Figure 8 and associated text: should base D not pair with base S via two H-bonds? (compare bottom left and bottom right base pair configurations). If there indeed is a single H-bond, an explanation and a reference are in order.

4) Discussion: This could do with some revision and made shorter and 'crisper', so that it is not largely a restatement of results. Do substantive insights emerge after carefully examining the structural changes alluded to below? For example, there may be a missed opportunity to compare the current structure with 4U7B, that is a pre-NTS-cleavage state (Dornan et al., 2015, NAR 43, 2424). I think readers would be interested in how the protein conformation has changed to accommodate what is now the target strand. While I understand that visualization of conformational changes using static images is hard to do well, I can imagine that a morphing movie could be informative. This can be easily done in these days, as a variety of tools are available to accomplish this. I think it would be good to point out for the less initiated that the NTS used is the authentic product of the first cleavage. It would be also good to define the T_0_ phosphate as the scissile phosphate (which I believe it is).

The authors might also have commented on [and cited] the bacterial IS630 family of elements that superficially have a related transposition and integration site selection mechanism.

Furthermore, the implications of the results for transposition/integration in the context of chromatin could be discussed in more detail. Mos1 is a eukaryotic transposon and has to deal with nucleosomes in some way, and recent evidence indicates that Tc1/mariner transposition avoids nucleosomes (see PubMed-ID 26755332). This ties up beautifully with the current data: a 150° kink would be incompatible with nucleosomal structure. Indeed, even in the case of PFV intasome, which induces a 60° bend in target DNA, engagement of a nucleosome requires peeling of the DNA from the histone octamer (Maskell et al., 2015). The last sentences of the Discussion are rather contrived and weak; something more insightful could be stated, or perhaps deleted [they do nothing to enhance the otherwise excellent presentation].

---

## [Author Response]

Essential revisions:

Below the comments that you need to take into account in your revision are listed. Also note that eLife expects papers to refer to the organism system in either the title or Abstract so I would propose you add a clause in the Abstract stating briefly that Mos1 is derived from Drosophila mauritiana.

We have added this clause to the second sentence of the Abstract.

I should add that all reviewers were enthusiastic about this work, and their view is encapsulated by the statement of one reviewer: 'This paper by Dr. Richardson and co-workers is impressive overall, as it provides novel and substantial insights to Mos1 and target DNA interactions. The experimental strategy to obtain crystals is clever and clearly worked out very nicely. I found the biochemical work, including the fluorescence spectroscopy, persuasive. When taken together, the correlation of the structural results with the solution biochemical data brings this work to a very high standard'.

*1) It would help to show how similar the STC is to the previously published transpososome (PEC) structures. A superposition (and/or rmsd values) would be very useful.*

We have added a section comparing the STC to our previous structures to the Discussion, along with an associated new Figure 9. See also our response to point 4 below.

Importantly, the molecular replacement was solved with the PEC complex, 3HOS, as the search model; one reviewer recalls that 3HOS has a peculiar packing that was proposed to mimic target DNA. Whether this proposal is still valid or not, the reviewer assumes that the TS strand must have been built in with some kind of difference electron density after molecular replacement found the orientation and the position of the complex. This difference electron density (before any model building) should be shown (at the very least as a supplementary figure) as it is crucial for everything in the paper-this should be explicitly shown.

The molecular replacement was performed with 3HOS (chains A to F only, corresponding to the transposase and the IR DNA); we did not include the DNA chains G and H, which take part in the “peculiar packing” that the reviewer describes. We have clarified this in the Materials and methods. As requested, we have added a supplemental figure (Figure 2—figure supplement 1) showing the Fo-Fc difference electron density map (at 2.3σ) after molecular replacement and before building of the target DNA.

Furthermore, as the resolution is moderate, possible model bias is a concern. The reviewer believes that the electron densities shown are inappropriate, as they are the 2Fo-Fc kind, and therefore subject of the modeling assumptions. In the reviewer's opinion, all the electron density figures should be replaced with simulated annealed omit maps (easily and rapidly computed with the freely available Phenix suite of programs).

We now show a simulated annealing composite omit map in Figure 2 and Figure 8.

Considering the similarities of the STC to the previously published PEC structures it is hard to imagine that target DNA would bend after strand transfer. Would a straight target DNA fit between active sites of the Mos1transpososome?

We don’t think so, and this is now included in the new section of the Discussion (first paragraph).

2) Discussion, seventh paragraph: Contrary to the oft-repeated dogma, the ability/propensity of retroviruses to integrate into nucleosomes is not due to the "pre-bent" nature of nucleosomal DNA (which is relatively smooth), but more likely due to complex interactions of the intasome with both gyres of nucleosomal DNA and the histone octamer. The interactions collectively offset the energetic penalty associated with the much more severe bending of the nucleosomal DNA between active sites of the intasome. Related: the second paragraph of the Introduction seems to suggest that nucleosomal DNA is not B-form.

Discussion section accordingly. This now reads:

“In the PFV integrase target capture and strand transfer complexes (Maertens et al., 2010) naked target DNA is bent by 55°. Nucleosomal DNA is peeled from the histone octamer and similarly deformed by interactions with the PFV intasome, providing a structural basis for retroviral integration at nucleosomes (Maskell et al., 2015).”

Similarly the Introduction now reads:

“Some retroviral integrases (e.g. prototype foamy virus (PFV) and HIV-1) preferentially insert their viral genome into nucleosomal DNA (Pruss et al., 1994, Maskell et al., 2015).”

3) Figure 8 and associated text: should base D not pair with base S via two H-bonds? (compare bottom left and bottom right base pair configurations). If there indeed is a single H-bond, an explanation and a reference are in order.

The modified base D is proposed to pair with S via one H-bond, between the 6-amino group of di-amino purine and the carbonyl oxygen of 2-thiothymine. This was described by IV Kutyavin et al. in Biochemistry (1996), 35, pp 11170-11176 and we have added this reference. They stated that the base pair acts like a mismatch; a steric clash between the 2-thio group of S and the 2-amino group of D tilts the bases relative to each other, precluding formation of the expected second H-bond. We have amended Figure 8 accordingly, and added this explanation to the figure legend.

4) Discussion: This could do with some revision and made shorter and 'crisper', so that it is not largely a restatement of results.

We have revised the Discussion significantly. We have removed much of the restatement of results, added a comparison of the STC with previous structures and included discussion of the implications for transposon integration in the context of chromatin. As a result, the Discussion is now slightly shorter.

Do substantive insights emerge after carefully examining the structural changes alluded to below? For example, there may be a missed opportunity to compare the current structure with 4U7B, that is a pre-NTS-cleavage state (Dornan et al., 2015, NAR 43, 2424). I think readers would be interested in how the protein conformation has changed to accommodate what is now the target strand. While I understand that visualization of conformational changes using static images is hard to do well, I can imagine that a morphing movie could be informative. This can be easily done in these days, as a variety of tools are available to accomplish this.

We have added comparisons of the STC with the *pre*-TS cleavage PEC (4U7B) and the post-cleavage PEC (3HOS) in the first paragraph of the Discussion and a new Figure 9 showing static comparisons. Differences in the orientation and position of T_0_ in the Mos1 STC compared to the thymine (T_57_) of the flanking target site duplication in 4U7B (Figure 9) support deformation of target DNA upon capture. By contrast, T_0_ closely aligns with T_54_ of the additional DNA duplex in 3HOS (Figure 9), which may represent the target strand before integration. Changes in the transposase structure are subtle and involve movement of the catalytic domains towards each other and the target DNA. We have made morphing movies showing how the Mos1 transposase conformation changes between the *pre*-TS cleavage PEC (4U7B) or the post-cleavage PEC (3HOS) and the strand transfer complex. These are included as Video 1 and Video 2, respectively.

I think it would be good to point out for the less initiated that the NTS used is the authentic product of the first cleavage.

We have amended the description of the DNA sequences in the Methods section (subsection “Preparation of ds DNA substrates”, first paragraph) to make this clearer.

It would be also good to define the T_0_ phosphate as the scissile phosphate (which I believe it is).

This has now been defined explicitly in the first paragraph of the subsection “Strand transfer products are in proximity to the active sites”.

The authors might also have commented on [and cited] the bacterial IS630 family of elements that superficially have a related transposition and integration site selection mechanism.

We have now mentioned IS630 elements in the Introduction and referenced a recent review of the superfamily be Tellier et al.

Furthermore, the implications of the results for transposition/integration in the context of chromatin could be discussed in more detail. Mos1 is a eukaryotic transposon and has to deal with nucleosomes in some way, and recent evidence indicates that Tc1/mariner transposition avoids nucleosomes (see PubMed-ID 26755332). This ties up beautifully with the current data: a 150° kink would be incompatible with nucleosomal structure. Indeed, even in the case of PFV intasome, which induces a 60° bend in target DNA, engagement of a nucleosome requires peeling of the DNA from the histone octamer (Maskell et al., 2015).

Thank you for pointing this out. We have extended the second last paragraph of the Discussion to include these points and the reference.

The last sentences of the Discussion are rather contrived and weak; something more insightful could be stated, or perhaps deleted [they do nothing to enhance the otherwise excellent presentation].

We have deleted the final two sentences.